# Barriers and facilitators of use of analytics for strategic health and care decision-making: a qualitative study of senior health and care leaders' perspectives

Elizabeth Ingram [iD],[1] Silvie Cooper,[1] Sarah Beardon,[1] Katherine Körner,[2] Helen I McDonald [iD],[3] Sue Hogarth,[4] Manuel Gomes [iD],[1] Jessica Sheringham [iD] [1]

¹Department of Applied Health Research, University College London, London, UK
²Royal Free London NHS Foundation Trust, London, UK
³Department of Infectious Disease Epidemiology, London School of Hygiene and Tropical Medicine, London, UK
⁴London Boroughs of Camden and Islington, London, UK

**Correspondence to**
Elizabeth Ingram;
e.ingram.17@ucl.ac.uk

## ABSTRACT

**Objective** This study investigated the barriers and facilitators that senior leaders' experience when using knowledge generated from the analysis of administrative health or care records ('analytics') to inform strategic health and care decision-making.

**Setting** One London-based sustainability and transformation partnership (STP) in England, as it was on the cusp of forming an integrated care system (ICS).

**Participants** 20 senior leaders, including health and social care commissioners, public health leads and health providers. Participants were eligible for inclusion if they were a senior leader of a constituent organisation of the STP and involved in using analytics to make decisions for their own organisations or health and care systems.

**Design** Semi-structured interviews conducted between January 2020 and March 2020 and analysed using the framework method to generate common themes.

**Results** Organisational fragmentation hindered use of analytics by creating siloed data systems, barriers to data sharing and different organisational priorities. Where trusted and collaborative relationships existed between leaders and analysts, organisational barriers were circumvented and access to and support for analytics facilitated. Trusted and collaborative relationships between individual leaders of different organisations also aided cross-organisational priority setting, which was a key facilitator of strategic health and care decision-making and use of analytics. Data linked across health and care settings were viewed as an enabler of use of analytics for decision-making, while concerns around data quality often stopped analytics use as a part of decision-making, with participants relying more so on expert opinion or intuition.

**Conclusions** The UK Governments' 2021 White Paper set out aspirations for data to transform care. While necessary, policy changes to facilitate data sharing across organisations will be insufficient to realise this aim. Better integration of organisations with aligned priorities could support and sustain cross-organisational relationships between leaders and analysts, and leaders of different organisations, to facilitate use of analytics in decision-making.

## Strengths and limitations of this study

► A key strength of this work is that we have illustrated how leaders experience complex and wide-ranging barriers and facilitators of use of analytics for strategic decision-making at a time when areas were on the cusp of transitioning from local models of integration in England (sustainability and transformation partnerships (STPs)) to national statutory organisations (integrated care systems (ICSs)). Our findings are timely, as the use of data and analytics are viewed as central to the integration of services and integrated decision-making.

► Another strength is that we worked collaboratively and in partnership with a digitally engaged and innovative site to inform the study design, research questions, study materials and study procedures.

► We recruited participants from a wide range of roles and constituent organisations of the study site, offering a breadth of perspectives.

► A limitation is that we recruited from one London-based STP (now ICS) and, while we believe most findings are transferable to other settings, all findings may not be transferable to settings that are perhaps less digitally engaged or have different priorities.

► We interviewed participants between January 2020 and March 2020 before the onset of the COVID-19 pandemic, which may have changed leaders' use of analytics for strategic health and care decision-making as well as the barriers and facilitators senior leaders' face when using analytics in this context.

## INTRODUCTION

Over the past 10 years, health and care reforms in England have been moving toward greater integration between different organisations concerned with the provision, commissioning and planning of health and care.[1–4] In England, care services include social care, which provides support to those with illness and/or disability with their activities of daily life. As part of reforms, all areas in England

were statutorily required to form integrated care systems (ICSs) by April 2021, replacing pre-existing sustainability and transformation partnerships (STPs). STPs and ICSs are place-based partnerships between local national health service (NHS) organisations, local authorities and other strategic partners with the intention of pooling resources to coordinate health and care services.

As health and care organisations move toward greater integration, senior leaders are increasingly required to make decisions about the structure and delivery of services (strategic decisions) that can have implications across organisational and sectoral boundaries. In England, the use of knowledge generated from the analysis of administrative data ('analytics') is seen as central to integrated decision-making and viewed as an opportunity to address health inequalities and the rising challenge of multiple long-term conditions. For example, a recent government White Paper states that 'integrating care … relies on the power of digital and data to join up care and uses that power to drive transformation of care'.[4] While there are many ways in which care may be integrated, analytics may best contribute to elements of organisational integration (the integration of formal organisational structures) and functional integration (the integration of back-office functions), as described in Mowlem and Fulop's framework.[5] To this end, analytics can aid assessments of local need to support development of new, more integrated services or used to monitor the effectiveness, efficiency and quality of existing services.[4 6–9]

Operational barriers to generating high-quality analytics have been well described, as have barriers to evidence-based decision-making in the NHS and for public health.[9–17] Barriers to evidence-based decision-making in these contexts include lack collaborative working relationships between leaders of different organisations, poor relationships between evidence producers and users, and competing or different organisational priorities.[9 11 17] However, less focus has been paid to the relational aspects of accessing and using analytics for strategic decision-making and little attention has been paid to senior leaders' readiness to use analytics, with findings suggesting leaders do not always value and use analytics for decision-making.[6 9 14] Furthermore, no previous studies have examined barriers and facilitators of use of analytics for strategic decision-making that has implications across health and care organisational and sectoral boundaries (hereafter 'strategic health and care decision-making'). Elucidating this understanding is important to help realise the White Papers' aims for data to transform care. This study investigated the barriers and facilitators that senior leaders' experience when using analytics for strategic health and care decision-making. A single STP was chosen as a case study to give nuanced, empirically rich and context-specific findings.

## METHODS

We conducted a case study of one STP in London, England, prior to its formation of an ICS.[18] This STP expressed interest in understanding barriers and opportunities to enhance senior leaders' use of analytics and was a site actively pursuing linkage of health and local authority records. It included participants from Clinical Commissioning Group (CCGs), local authorities, hospitals and other service providers. Figure 1 presents an overview of stakeholders in this case study. This manuscript was prepared following the Standards for Reporting Qualitative Research (SRQR) Checklist.[19]

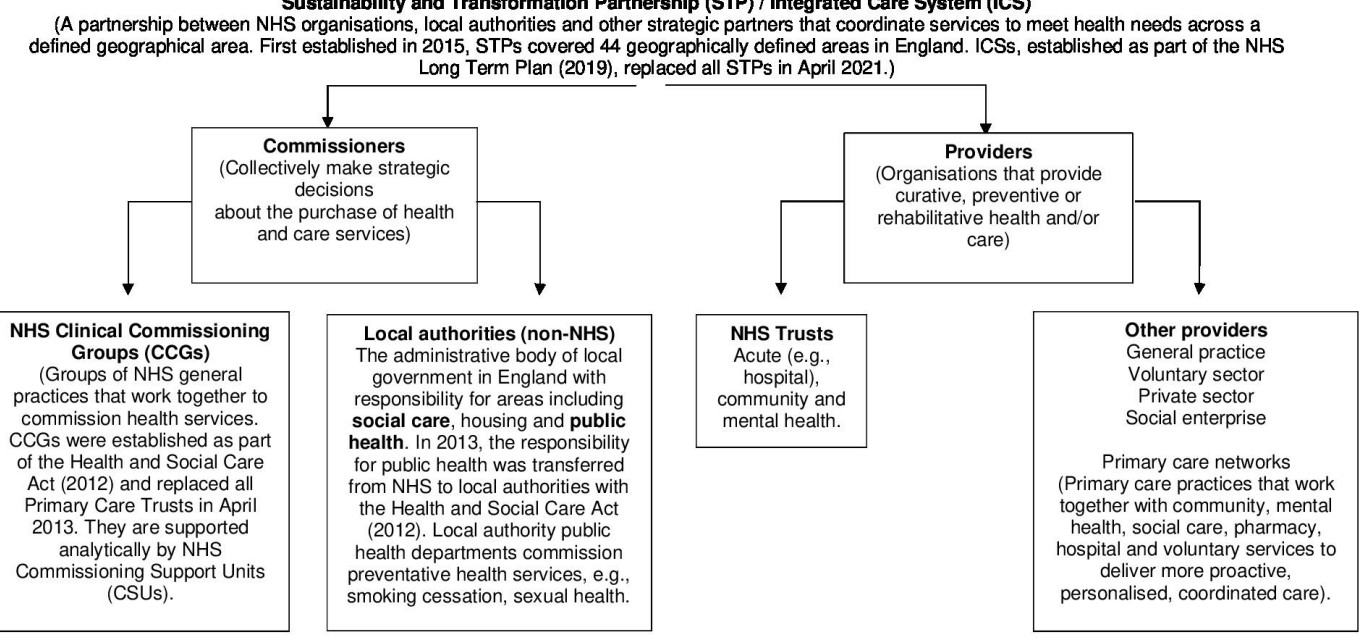

**Sustainability and Transformation Partnership (STP) / Integrated Care System (ICS)**
(A partnership between NHS organisations, local authorities and other strategic partners that coordinate services to meet health needs across a defined geographical area. First established in 2015, STPs covered 44 geographically defined areas in England. ICSs, established as part of the NHS Long Term Plan (2019), replaced all STPs in April 2021.)

**Commissioners**
(Collectively make strategic decisions about the purchase of health and care services)

**Providers**
(Organisations that provide curative, preventive or rehabilitative health and/or care)

**NHS Clinical Commissioning Groups (CCGs)**
(Groups of NHS general practices that work together to commission health services. CCGs were established as part of the Health and Social Care Act (2012) and replaced all Primary Care Trusts in April 2013. They are supported analytically by NHS Commissioning Support Units (CSUs).

**Local authorities (non-NHS)**
The administrative body of local government in England with responsibility for areas including **social care**, housing and **public health**. In 2013, the responsibility for public health was transferred from NHS to local authorities with the Health and Social Care Act (2012). Local authority public health departments commission preventative health services, e.g., smoking cessation, sexual health.

**NHS Trusts**
Acute (e.g., hospital), community and mental health.

**Other providers**
General practice
Voluntary sector
Private sector
Social enterprise

Primary care networks
(Primary care practices that work together with community, mental health, social care, pharmacy, hospital and voluntary services to deliver more proactive, personalised, coordinated care).

**Figure 1** The relationship between constituent organisations in the ICS interviewed this study, formerly called an STP. Figure adapted from The King's Fund explainer.[36] NHS, national health service.

## Recruitment

Participants were recruited from constituent organisations of the STP and eligible if responsible for strategic decision-making for their own organisation or local health and care system. Eligible participants were first identified and contacted by key STP leaders and then by the study team. Participants were asked to recommend further eligible colleagues.[20] Recruitment ended when we reached data saturation.

## Data collection

We conducted 20 semi-structured individual interviews between January 2020 and March 2020. Interviews followed a topic guide developed using guidance for conducting interviews in healthcare settings.[21] Participants were asked to describe their use of analytics as part of a strategic health and care decision they had made, and things that had facilitated or hindered their use. The guide was developed to reflect the STP's priorities and adapted to probe underexplored topics as the study progressed. Audio recordings of interviews were transcribed and anonymised by an external transcription agency and subsequently checked for accuracy. Once transcribed, recordings were deleted.

Transcripts were analysed using the framework method. This form of thematic analysis is suitable for multidisciplinary teams, where members vary in their experience of using qualitative analysis methods and want to use both inductive and deductive coding approaches to give a descriptive and holistic overview of the semi-structured interview data.[22] EI conducted the analysis by iteratively following steps from Braun and Clarke.[23] Codes were first generated deductively based on operational barriers to high-quality analytics previously identified in the literature.[13–15] Salient phrases were then coded inductively and subsequently compared with research questions. Codes were grouped to form categories and categories refined to represent a robust theme across participants. A reflexive journal was kept during interviews and referred to during analysis and write-up. A subset of transcripts were double coded by SC and SB, and the coding frame checked before being applied across the dataset.[22]

Research questions, the topic guide and study procedures were informed by a pilot study and refined prior to the full study.

## Participant engagement and involvement

The idea for this study was borne out of an expressed STP need to understand use of analytics for strategic decision-making from the perspective of senior leaders. Key staff at the STP reviewed the study protocol, topic guide, participant information and consent sheets. These materials reflected their priorities. Study materials were tested with a neighbouring STP site. We worked collaboratively with the STP throughout the research process and emergent themes were discussed during analysis and write-up. Our collaborators will choose how to disseminate study findings.

| Table 1 | Participant characteristics (N=20) |
|---|---|
| **Characteristic** | **N (%)** |
| Gender: male | 12 (60) |
| Geography | |
| Inner London borough | 8 (40) |
| Outer London borough | 4 (20) |
| Inner and outer London boroughs* | 8 (40) |
| Generic organisation and role | |
| Health: provider | 6 (30) |
| Health: commissioner | 4 (20) |
| Local authority: social care commissioner | 4 (20) |
| Local authority: public health consultant | 2 (10) |
| Health and local authority: health and social care commissioner | 4 (20) |

*Split role across inner and outer boroughs. Includes STP leads.
STP, sustainability and transformation partnership.

## RESULTS

Interviews were conducted with senior leaders in health and/or care commissioning, provider and public health roles (see table 1).

Participants described the process of attempting to obtain data and/or analytics for strategic health and care decision-making as uncoordinated, 'ad hoc' or 'random'. We found that factors related to three areas—individuals' working environments (theme 1), relationships (theme 2) and the quality of data sought (theme 3)—greatly influenced this process. These factors were barriers or facilitators of use of analytics depending on circumstances and contexts. They influenced if and how analytics were obtained and its utility for informing decision-making.

The purpose of analytics use for decision-making varied across the three themes. In most cases, analytics were used to monitor the quality or efficiency of existing services to improve care provision or justify investing or disinvesting in services. In other cases, analytics were used to better understand local needs to support the development of new services.

### Theme 1: working environments

Factors relating to individuals' working environments included organisational fragmentation and competing priorities and were described as barriers to analytics access and use.

### Organisational fragmentation

Participants worked across separate, fragmented health and care departments and organisations such as CCGs and local authorities. Those who recounted facing challenges when they had attempted to use analytics described how divisions between, and within, organisations created siloed data systems, which meant residents' records could be stored in different data systems if they contacted more than one service. At times, this made it difficult for leaders

to access data as they had to actively request data and/or analytics from individuals in other departments or organisations. Divisions in systems across organisations meant senior leaders did not always know who held certain data, whether the data they held would be relevant to inform decision-making or how to contact key individuals. These barriers were aptly described by one participant who had tried to access analytics to better understand and plan for social care accommodation needs:

> [RES]: We need housing data, we need social care data, we need some health data, but it's proving difficult to get those data sources … there's issues around [asking] "where does the data sit?". So, I had a meeting with [an internal team] asking for some data. They're like, "But this sits here, this doesn't sit with us". It's unclear who owns certain pieces of data and how best to extract it.
>
> [INT]: Is that the reason that you had issues accessing it in the first place?
>
> [RES]: Definitely. So, housing data, in particular, where it sits [is] in a completely different department, a different team. We have no right to access any of that data, so it will take quite a lot of time to get it. (ID023, social care commissioner)

For some, information governance requirements contributed to these barriers to data sharing across departmental and organisational boundaries. For example, General Data Protection Regulation, which is a legal framework for the collection and processing of personal data introduced in the UK in 2018, was described by one participant as follows:

> GDPR was supposed to make [sharing data safely] better or easier but I think that's caused a lot of complications as well … I think the trust [for me] is, [once I've shared my data with you] are you going to ensure that you're following the rules, so if there's a breach, I don't have to pay ten percent of my revenue. (ID015, health commissioner)

Overcoming barriers to data sharing often involved a time-consuming process, where participants had to identify who to request data from and justify their need. The former participant continued:

> Having to explain the rationale as to why we need data is always the start of it and can always be a bit of a challenge [in] trying to make them understand why I need access to this data and what it will be used for … . But I think the biggest thing is, everyone's busy … . it's never a priority when someone else comes saying, "Do you have this data source? I need it for X", because I think, "I've got twenty other things on my plate". (ID023, social care commissioner)

This time-consuming process requiring continuous justification was, therefore, described as an additional organisational barrier to data sharing, analytics access

and use of analytics—with other priorities and work often taking precedence. When participants could not access data held on siloed systems, some made decisions without all the 'necessary information' (ID022, health and social care commissioner), while others relied more on expert opinion (such as the opinion of single practitioners) or stopped their use of analytics.

### Competing priorities
Many participants described how fragmentation across their health and care system, at times, led to different or competing organisational priorities. In more extreme circumstances, this hindered collective priority setting for health and care decisions, despite organisations being encouraged to align priorities locally to facilitate collaboration. For instance, one participant expressed little motivation to engage in health and care decision-making and promote data sharing due to conflicting financial drivers:

> If we have a patient who we see in the hospital we get paid £70 or something for a follow-up. If we work out a new model of care where this patient can be seen in the community or virtually, we would get paid £10 or £15. What on earth would we want to do that for? … If you're saying let's [in a] wholesale [manner] move half of our patients into the community, let's lose all of that revenue, then suddenly the fixed costs that we have in this building and others become overwhelming. (ID011, health provider)

Indeed, several participants described how reservations around sharing data often stemmed from conflicting priorities. In addition, some participants stated they were more likely to share their data if they trusted that recipients had priorities aligned to their own and, as such, would use their data as they had specified. This was particularly relevant for data sharing between commissioners and providers, where providers were hesitant to share data in case commissioners used it to justify disinvestment in their services.

Interviewees also observed that they were often competing for analysts' time against the extensive mandatory requirements they faced from external public bodies such as NHS England:

> The structures that sit across us, there are data requirements placed on us which are often at short notice and unexpected or slightly different or very similar to one that we did previously. The time and energy and resources that it takes for [analysts] to keep changing that information and updating it and translating it into the latest format is time consuming, it's energy sapping … So, yeah it's not [the analysts'] priority to respond to our [analytics] requests immediately. (ID022, health and social care commissioner)

Externally mandated requirements that occurred frequently, unexpectedly and at short notice were, therefore, described as creating 'time consuming, energy sapping' work that needed to be prioritised over requests

from leaders for analytics support. This was described as a barrier to analytics access, which hindered leaders' use of analytics.

## Theme 2: individual relationships

Individual relationships between people involved in the process and decision—leaders and analysts, and leaders and leaders—were viewed as crucial. Participants described relationships as helping them overcome barriers stemming from organisational fragmentation.

### Leader–analyst relationships

Participants suggested that the uncoordinated way in which leaders obtained analytics meant relationships between leaders and analysts greatly influenced analytics access and use. Some leaders who experienced advanced use of analytics regularly collaborated with trusted analysts to obtain suitable analytics support. They described having a 'good dialogue' with analysts, which facilitated data access, and enabled leaders to iteratively and successfully review and use analytics to inform decisions. In explaining how a collaborative relationship with an analyst worked, one participant said:

> We kind of described the scope of the strategy, and what we'd intended it to do, and then [the analyst] went off and led [the work]. We had a couple of meetings to check in every so often … [the analyst] and I have worked together on and off for years … I just inherently trust [the analyst] to know what [they're] doing. (ID020, health commissioner)

The benefits of having 'good' working relationships with analysts appeared so crucial that leaders 'attach[ed] themselves to good analysts, even if external to their organisation:

> There's a better analyst in [an external organisation]. [And so,] I would nick [them] sometimes. I would trust [their] judgement around [how the analysis should be conducted]. (ID021, health and social care commissioner)

Participants who faced barriers when trying to make analytics-informed decisions typically stated that, while they wanted collaborative working relationships with analysts, these were not currently available. In some cases, some of these participants could not access data as they did not know who to contact. Those who could access data, but were reluctant to use analytics, described a struggle to develop questions that could be addressed without analysts' input. This led to 'insufficient' outputs, which did not address questions they required answering, lacked extra detail around how to interpret and use the output, or recommended unfeasible actions.

Organisational fragmentation was also described as creating physical disconnect between leaders and analysts, meaning that good, cross-organisational relationships were even more salient. For instance, one provider faced difficulties working with external analysts, as outputs did not contain details necessary for their decision-making. They felt this was because analysts were not 'part of the team' and, therefore 'didn't know what [the leaders were] talking about and leading on' with respect to a decision (ID012, health provider). This participant eventually hired an internal analyst to produce better-suited analytical support. Several participants believed that they had a better understanding of how services operated than analysts because analysts where not co-located in decision-making teams. This drove their choice to request raw data and conduct their own analyses to support decision-making, independent of analysts' input.

### Leader–leader relationships

Building trust and relationships between individual leaders of the organisations was also vital for some participants when making strategic decisions across organisational boundaries. More regular and confident users of analytics had established relationships and aligned strategic priorities with other, trusted leaders. Conversely, those who faced barriers to obtaining and using analytics from external organisations typically faced difficulties forming relationships with other leaders and aligning strategic priorities:

> We've got a new Director [of the partner organisation] come in, who very much sees that they've got to sort out this little corner [of the decision] as a separate project, rather than doing it all at once. Which has delayed the togetherness of the project.… We were talking 18 months ago, we'd got the model ready, and yet we're still sitting here now, talking about it. (ID016, health provider)

High turnover of senior leaders, in general, was also described as a barrier to developing and sustaining leader relationships, stalling project delivery and use of analytics.

## Theme 3: data quality

A third theme centres on data quality, which, when perceived as poor, was described as hindering senior leaders' use of analytics for strategic health and care decision-making. The term signified two issues: data availability and accuracy, and data linkage.

### Data availability and accuracy

Several participants described circumstances where data they required for a decision did not exist because certain groups had little or sporadic contact with services or recording of certain information was not mandatory. This hindered their ability to make decisions for these populations. For example, when discussing service provision for residents with autism, one commissioner stated that they 'simply don't know how many children have autism, because there are whole cohorts not recorded' (ID022, health and social care commissioner). They went on to describe how this made it difficult to accurately plan services, as they could not determine how may children had autism in the borough. Attempting to overcome this

issue, they retrospectively collected data, which was a resource-intensive and 'frustrating' task. They also relied on 'professional judgement', 'gut feeling' and academic studies 'carried out a long time ago' more so than analytics. This approach was common among participants who experienced data availability as a barrier.

Six participants described how concerns around data accuracy sometimes led to considerable resources being used to determine the 'correct' data, which stopped more advanced analytical work. In some cases, participants stopped their use of analytics as part of decision-making due to perceived data inaccuracies, and again relied more on expert opinion. More regular and confident analytics users rarely communicated data availability and accuracy as barriers to use of analytics.

### Data linkage
Participants describing concerns around data quality and difficulties they had faced accessing data because of siloed data systems or poor working relationships also reflected how projects that link patient records stored across data systems could help overcome these barriers. Without linkage, data were seen as being often disconnected and stored across siloed data systems that 'don't talk to each other'. For example, one commissioner described linking NHS and publicly available data on area-level deprivation to inform their decision-making in this example, prompting them to tailor services to different population groups:

> We looked at primary care data … then we looked at some acute data, and we managed to link the acute and primary care data. [After linking with publicly available deprivation data] what we ended up with was six very different projects, so not this blanket one size fits all. (ID013, health commissioner)

They described how this linked data enabled them to see the 'fuller picture' of service use for residents who accessed care across organisational boundaries. As a result, they felt more able to holistically understand health needs and more efficiently make strategic health and care decisions. However, they felt unable to make decisions that considered residents' individual social circumstances or social care use, as local authority records (containing such information) did not contain NHS numbers. NHS numbers were seen as necessary enablers of data linkage:

> [With] our local authority data, unfortunately, they didn't use NHS number at all. So normally where you might get say a 65% to 70% match, or even a 50%/60% … we had nothing … the local authority data could have added value. (ID013, health commissioner)

This participant was fairly exceptional as they conducted their own linkage, and other participants did not currently have access to data linked across services. Most participants expressed a positive view of the potential for data linkage to help them understand needs

and inform strategic health and care decision-making. Without linked data, participants made decisions with incomplete data that were 'heavily caveated' and evaluated, or again sought alternative information. A handful of participants were setting up data systems that linked records across health and care organisations to enable leaders' access to linked data.

## DISCUSSION
In this qualitative study, we found that senior leaders' use of analytics for strategic health and care decision-making was influenced by the degree and nature of connectedness between organisations, individuals and data.

### Improving organisational integration and strengthening relationships between leaders and analysts should enable leaders to better use data to transform care
At the time of interviews, constituent STP organisations were structurally independent. This hindered analytics access and use by creating siloed data systems, which consistently create barriers to health and care integration in the UK.[24 25] As a result, most participants could not follow patient or resident journeys across services, nor plan services effectively using data that might be linked across this journey. Our findings support Mowlan and Fulop's framework by suggesting that greater use of analytics for decision-making may help achieve increased organisational and functional integration. Our findings also suggest that increased integration at organisational and functional levels through joined up data systems could facilitate the use of analytics for informing strategic health and care decision-making.[5]

In March 2020, sharing of certain data across organisational boundaries was mandated to support the UK COVID-19 response. This demonstrated that improved data sharing across health and care is possible and important for care delivery, with the governments' 2021 White Paper legislating reforms aiming to continue increased data sharing.[4] Linking data across organisational boundaries is also viewed as a potential enabler of more integrated care.[2 7 13] However, the White Paper did not discuss data linkage, instead generally committing to improving data availability and quality.[4] Our findings suggest that programmes linking administrative data across health and care are welcomed and, if successful, could help improve care delivery.[2 26] We found that, when data were linked across primary and acute care, one participant felt better able to understand needs and tailor commissioned services. However, they faced difficulties understanding wider determinants of health that would require local authority data. It is unclear how upcoming reforms propose to improve data sharing with local government.[27] It is crucial that the national government's forthcoming data strategy for health and care considers how to improve data sharing with local government, which could facilitate health and care integration and help realise aims to tackle health inequalities.[28]

Our findings suggest that, while necessary, these data-related policy changes alone will be insufficient to realise the White Paper's aspiration for data to drive the transformation of care.[4] When these reforms come into force, leaders may continue to struggle accessing and using data and/or analytics if they do not know where different data are held, who to contact to request certain data or believe analysts do not understand decision-making contexts. This aligns with previous literature highlighting how NHS leaders with different professional backgrounds can differ in their use of evidence for decision-making and literature emphasising how relationships between evidence producers and users can influence evidence use in UK public health decision-making.[9 10] Following reforms, leaders may also continue to distrust the quality of data, which has also been identified as a concern in the previous literature.[15 25 29] We found that leaders with working relationships with trusted analysts were able to overcome these barriers and work collaboratively to obtain analytical support. Efforts to develop and sustain relationships between leaders and analysts across organisations are, therefore, crucial. These could include analyst secondments that provide analysts greater proximity to decision-makers and foster shared understanding of values and decision-making contexts.

While the 2021 White Paper reforms include changes aiming to facilitate shared priority setting across organisational boundaries, separate financial budgets will remain for NHS and local government.[4] This is concerning as we found that financial structures continue to disincentivise cross-sectoral working, particularly in hospital settings where investments in system-wide priorities can conflict with the priorities of individual organisations.[30 31] Fundamental changes in financial incentives are needed to ensure alignment of strategic priorities across health and care, particularly if shared priority setting is viewed as a cornerstone of integration.[4] We found that good working relationships between leaders of different organisations circumvented organisational barriers by facilitating shared priority setting. However, intense resources were required to develop and sustain these relationships, with high staff turnover stalling the progress and delivery of cross-organisational programmes of work, as seen previously.[7 24 32] Where these relationships were absent, strategic priorities were misaligned and at times conflicting, which significantly hindered health and care decision-making. These findings align with previous literature, which reports leader–leader relationships as one of the most important predictors of successful and sustainable partnership working in health and care, as well as a key determinant of evidence use in NHS and public health decision-making.[7 9 11 12 15 17 24 33 34]

### Strengths and limitations of this study

There is little peer-reviewed literature on the use of analytics by senior leaders for joint decision-making. While we have identified familiar factors that continue to facilitate and hinder integration, this study offers novel and rich insights into the complexity of barriers and facilitators of use of analytics for strategic decision-making when areas were on the cusp of transitioning from local models of integration (STPs) to statutory organisations (ICSs). Furthermore, we show how these experiences can impact decision-making. Participants were from a wide range of roles and organisations, offering a breadth of perspectives.

We recruited from one London-based STP with digitally engaged leadership that, during recruitment, were actively pursuing a data linkage programme to facilitate formation of an ICS. Therefore, all of our findings may not be transferable to other settings.[35] Despite the STPs' overall relative digital innovation, we still identified extensive barriers to use of analytics and there remained considerable variation in interest in data across the STP. It is likely that these barriers, plus others, are more impactful in less digitally engaged ICSs. In addition, sharing of certain data across organisations was mandated as part of the UK COVID-19 response. Barriers related to data sharing may, therefore, not be relevant in times of crisis but remain important for future partnership working and provide insight into possible strategies that could facilitate use of analytics.

### Implications for policy and practice

To realise the White Paper's aspiration for data as a driving force for health and care integration, more is needed to better integrate organisations, align organisational priorities and build and sustain cross-organisational relationships between leaders and analysts, and leaders of different organisations. While policy changes to facilitate data sharing across organisations are necessary, they will be insufficient without strategies to address these further key barriers to use of analytics for strategic health and care decision-making.

**Contributors** EI, HIM, MG, SH and JS conceptualised the study and devised the research questions and methods. EI applied for the ethical approval and conducted all interviews. EI and SB analysed interview data with the support of SC, KK and JS. EI drafted the manuscript and acts as guarantor of the work. All authors commented on drafts of the manuscript and agreed the decision to submit for publication.

**Funding** This study is an independent research funded by the National Institute for Health Research School for Public Health Research (grant reference number: PD-SPH-2015-10025) and the National Institute for Health Research Applied Research Collaboration (ARC) North Thames. The views expressed in this publication are those of the authors and not necessarily those of the National Institute for Health Research or the Department of Health and Social Care.

**Competing interests** SH is employed by and JS is seconded for part of her time in one of the organisations within the STP case study site.

**Patient and public involvement** Patients and/or the public were involved in the design, or conduct, or reporting, or dissemination plans of this research. Refer to the Methods section for further details.

**Patient consent for publication** Not applicable.

**Ethics approval** This study involves human participants and received ethical approval from University College London's Research Ethics Committee (reference number: 15847/001). All those participating in the study gave their informed consent before taking part.

**Provenance and peer review** Not commissioned; externally peer reviewed.

**Data availability statement** No data are available. Sharing of data outside the research team was not approved as part of the necessary ethical approvals for this study.

**ORCID iDs**
Elizabeth Ingram http://orcid.org/0000-0002-0354-4551
Helen I McDonald http://orcid.org/0000-0003-0576-2015
Manuel Gomes http://orcid.org/0000-0002-1428-1232
Jessica Sheringham http://orcid.org/0000-0003-3468-129X

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
