## [Reviewer comments · BMJ Open]

ARTICLE DETAILS

TITLE (PROVISIONAL)	Barriers and facilitators of use of analytics for strategic health and care decision-making: a qualitative study of senior health and care leaders' perspectives
AUTHORS	Ingram, Elizabeth; Cooper, Silvie; Beardon, Sarah; Körner, Katherine; McDonald, Helen; Hogarth, Sue; Gomes, Manuel; Sheringham, Jessica

VERSION 1 – REVIEW

REVIEWER	Jones, Lorelei Bangor University
REVIEW RETURNED	15-Oct-2021

GENERAL COMMENTS	Thank you for the opportunity to read this paper. It is clearly written and the introduction provides helpful background information on the research setting. The methods and sample are appropriate and the findings provide useful insights for a policy audience. The topic is a significant and long-standing issue for practitioners involved in implementing integrated care strategies but I have not seen a lot of research in this area. I suggest only minor revisions as follows.  1. You note that leaders may use other forms of knowledge, such as experience, expertise or academic papers – might this be an accomplishment in the absence of perfect data (which is inevitable in complex, resource-constrained systems)? 2. Consider locating your findings in relation to research by Davide Nicolini on how CEOs use knowledge, there might be something that resonates (or contrasts) with your research, or it might be helpful in your background and rationale in terms of identifying a 'gap' in what is known about how leaders use knowledge. 3. Consider adding a sentence to your background that explicates the expected benefits of data analysis beyond 'driving' integrated care. Suggestions for minor edits: Page 5, L 1, the pronoun 'we' is surprising in an HSR paper, it suggests the researchers are positioning themselves as healthcare managers, consider rephrasing, e.g. 'As health and social care organisations move towards ...' P5, L29 The final two 'hanging' sentences can be added to the paragraph above. Figure one – it might be better to provide an overview of your case study rather than a 'typical' partnership In the 'commissioners' box – I suggest removing the word 'individuals' to reflect the fact that commissioning is collective decision-making
--

	In the 'Local Authorities' box – the following sentence is not clear 'Public health commission preventative health services, e.g., smoking cessation, sexual health'. When you say 'public health' are you referring to some sort of collective body e.g a department/organisation? P8, L15. I suggest changing 'medical research' to 'healthcare settings' otherwise your reader may wonder why you used guidance on developing a topic guide for medical research when you are doing organisational research. P8, L19 'anonymised' rather than 'deidentified' P8, L 25 'A suitable' rather than 'the most suitable'. P8, L 25 I know what you mean when you say 'when candidate themes have been identified before the data are analysed' however this may be misinterpreted by some readers. I suggest revisiting the Gale et al paper and editing slightly this section on the benefits of the Framework Method for policy-relevant research and multi-disciplinary research teams. P 8, L31 Throughout the paper consider writing 'use of data analysis' or 'use of analytics' rather than 'analytics use' P10, L16 'For some, information governance requirements contributed to these barriers to data sharing across departmental and organisational boundaries'. Can you elaborate and give an example to illustrate? P 10, L39 is there an alternative to 'halted'? e.g 'stopped' or 'gave up' Specifying 'Theme 1' and 'Sub-theme 1' etc is unnecessary for your reader and a bit stilted. You could remove these and just use the headings and sub headings P121, L 7 Perhaps use [external organisation] instead of XXX P12, L19 The writing here is too condensed and the list structure of this sentence detracts from the rich insights. Consider rewriting, perhaps expanding into more than one sentence. P12, L48 The data extract is difficult to read and doesn't really add very much. Consider removing.
--	--

REVIEWER	Tweed, Emily University of Glasgow, MRC/CSO Social and Public Health Sciences Unit
REVIEW RETURNED	25-Oct-2021

GENERAL COMMENTS	This paper describes an interesting study that is timely and well conducted. I have one major point of feedback and several minor points. The major point of feedback is that I felt the paper would benefit from referencing the broader evidence base about how evidence is used in decision-making in health and social care, as many of the themes highlighted in relation to analytics are common to this broader literature. I also felt it might be worth revisiting in the discussion the theoretical framework about integration mentioned in the introduction, in order to discuss the findings in that context. Minor points:  - there is a minor typo in objectives on p3 - the sentence does not read correctly, is there a word missing?
---

	- the bullet points under "strengths and limitations" would benefit from being revised slightly. The first bullet simply describes what was done and doesn't clearly articulate the specific strength - is it the timing in relation to ICS/STPs? If so this needs to be more the focus of this sentence. Can this be phrased in terms that make it clearer how this strength makes these findings useful to others? The second point mentions the focus of the study as a digitally engaged site - this is potentially a strength or a limitation; is the strength being flagged here instead the partnership working to inform the study design and conduct? - in the introduction, the phrase "health and care" is used frequently; I think it would be worthwhile defining what care means in this context, especially for international readers e.g. social care that provides support of people with activities of daily living etc. Other than these points I think this article is an interesting contribution to the literature and makes some salient points in relation to the current UK policy context.
--	---

VERSION 1 – AUTHOR RESPONSE

Thank you for the opportunity to revise this manuscript and for the very constructive set of comments from reviewers. In the table below, we provide point by point responses to each comment. The additions needed to sufficiently address reviewers' comments have led to the word count for the manuscript being over 4000 words. The corresponding changes to the manuscript have been made using tracked changes (Manuscript_marked copy).

Reviewer 1 Comments	Our responses	Page number and line
You note that leaders may use other forms of knowledge, such as experience, expertise or academic papers – might this be an accomplishment in the absence of perfect data (which is inevitable in complex, resource-constrained systems)?	We agree that use of other forms of knowledge in the absence of perfect data is certainly an accomplishment. We've included this information in our manuscript to demonstrate that participants often either sought other forms of knowledge or chose to stop their use of analytics when faced with certain barriers, and to highlight how leaders differed in their responses to certain barriers.	
Consider locating your findings in relation to research by Davide Nicolini on how CEOs use knowledge, there might be something that resonates (or contrasts) with your research, or it might be helpful in your background and rationale in terms of identifying a 'gap' in what is known about how leaders use knowledge.	We have made changes to the introduction of the manuscript to refer to the wider literature on how evidence is used in decision-making in health and social care to describe how our study fits with these gaps. We have added the following: "Operational barriers to generating high-quality analytics have been well described, as have barriers to evidence-based	P5 L25

	decision-making in the NHS and for public health⁹⁻¹⁸. Barriers to evidence-based decision-making in these contexts include lack collaborative working relationships between leaders of different organisations, poor relationships between evidence producers and users, and competing or different organisational priorities^{9,11,12,18}. However, less focus has been paid to the relational aspects of accessing and using analytics for strategic decision-making and little attention has been paid to senior leaders' readiness to use analytics..." We have also made changes to the discussion of this manuscript to position our findings within this wider literature. When discussing relationships between leaders and analysts, we have added the following: "This aligns with previous literature highlighting how NHS leaders with different professional backgrounds can differ in their use of evidence for decision-making and literature emphasising how relationships between evidence producers and users can influence evidence use in UK public health decision-making". When discussing relationships between leaders of different organisations, we have added the following: "These findings align with previous literature, which reports leader-leader relationships as one of the most important predictors of successful and sustainable partnership working in health and care, as well as a key determinant of evidence-use in NHS and public health decision-making" (see tracked changes).	P14 L20 P15 L5
Consider adding a sentence to your background that explicates the expected benefits of data analysis beyond 'driving' integrated care.	We've added the following to the second paragraph to further develop our points here: "the use of knowledge generated from the analysis of administrative data ('analytics') is seen as central to integrated decision-making and viewed as an opportunity to address health inequalities and the	P5 L16

	rising challenge of multiple long-term conditions."	
Figure one – it might be better to provide an overview of your case study rather than a 'typical' partnership In the 'commissioners' box – I suggest removing the word 'individuals' to reflect the fact that commissioning is collective decision-making In the 'Local Authorities' box – the following sentence is not clear 'Public health commission preventative health services, e.g., smoking cessation, sexual health'. When you say 'public health' are you referring to some sort of collective body e.g a department/organisation?	Figure 1 has been changed to reflect the stakeholders from our case study site specifically. The title of this figure has been changed to "An overview of partners within our case study site (a single NHS Integrated Care System in London, England) at the time of interviews" to reflect this. Commissioners are now described as "collectively" making strategic decisions, and clarity around the description of public health departments has been added (e.g., "Local authority public health departments commission..")	Additional document
P8, L 25 I know what you mean when you say 'when candidate themes have been identified before the data are analysed' however this may be misinterpreted by some readers. I suggest revisiting the Gale et al paper and editing slightly this section on the benefits of the Framework Method for policy-relevant research and multi-disciplinary research teams.	We have edited this section to better describe the benefits of using the Framework Method in this context: "Transcripts were analysed using the Framework Method. This form of thematic analysis is suitable for multi-disciplinary teams where members vary in their experience of using qualitative analysis methods and want to use both inductive and deductive coding approaches to give a descriptive and holistic overview of the semi-structured interview data."	P6 L27
P10, L16 'For some, information governance requirements contributed to these barriers to data sharing across departmental and organisational boundaries'. Can you elaborate and give an example to illustrate?	The following has been added: "For example, General Data Protection Regulations (GDPR), which is a legal framework for the collection and processing of personal data that was introduced in the UK in 2018, was described by one participant as follows: "GDPR was supposed to make [sharing data safely] better or easier but I think that's caused a lot of complications as well...I think the trust [for me] is, are you going to ensure that you're following the rules, so if there's a breach, I don't have to pay ten percent of my revenue". (ID015, Health Commissioner)"	P8 L31
P12, L19 The writing here is too condensed and the list structure of	We have restructured this section slightly to make it easier to read and	P12 L10

this sentence detracts from the rich insights. Consider rewriting, perhaps expanding into more than one sentence.	ensure our key points come across. We have made the following edit: “For example, when discussing service provision for residents with autism, one commissioner stated they “simply don’t know how many children have autism, because there are whole cohorts not recorded” (ID022, Health and Social Care Commissioner). They went on to describe how this made it difficult to accurately plan services, as they could not determine how many children had autism in the borough.”	
P12, L48 The data extract is difficult to read and doesn’t really add very much. Consider removing.	Given that linking data across organisational boundaries is seen as a potential enabler of more integrated care, we feel that retaining this extract illustrates the opportunities to address inequalities and tailor services that linking data affords. We have, however, edited the section to make it easier to read: “For example, one commissioner described linking NHS and publicly available data on area-level deprivation to inform their decision-making in this example, prompting them to tailor services to different population groups: “We looked at primary care data... then we looked at some acute data, and we managed to link the acute and primary care data. [After linking with publicly available deprivation data] what we ended up with was six very different projects, so not this blanket one size fits all.” (ID013, Health Commissioner)”	P12 L31
Minor formatting comments:  • Page 5, L 1, the pronoun ‘we’ is surprising in an HSR paper, it suggests the researchers are positioning themselves as healthcare managers, consider rephrasing, e.g. ‘As health and social care organisations move towards ...’ • P5, L29 The final two ‘hanging’ sentences can be added to the paragraph above. 	All these minor comments on formatting of our manuscript/ suggesting slight changes to wording have been addressed, as advised by the reviewer (see manuscript with tracked changes).	Throughout document

 • P8, L15. I suggest changing 'medical research' to 'healthcare settings' • P8, L19 'anonymised' rather than 'deidentified' • P8, L 25 'A suitable' rather than 'the most suitable'. • P 8, L31 Throughout the paper consider writing 'use of data analysis' or 'use of analytics' rather than 'analytics use' • P 10, L39 is there an alternative to 'halted'? e.g 'stopped' or 'gave up' • Specifying 'Theme 1' and 'Sub-theme 1' etc is unnecessary for your reader and a bit stilted. You could remove these and just use the headings and sub headings • P12, L 7 Perhaps use [external organisation] instead of XXX 		
Reviewer 2 Comments		
The major point of feedback is that I felt the paper would benefit from referencing the broader evidence base about how evidence is used in decision-making in health and social care, as many of the themes highlighted in relation to analytics are common to this broader literature.	We have made changes to the introduction of the manuscript to refer to the wider literature on how evidence is used in decision-making in health and social care to describe how our study fits with these gaps. We have added the following: "Operational barriers to generating high-quality analytics have been well described, as have barriers to evidence-based decision-making in the NHS and for public health⁹⁻¹⁸. Barriers to evidence-based decision-making in these contexts include lack collaborative working relationships between leaders of different organisations, poor relationships between evidence producers and users, and competing or different organisational priorities^{9,11,12,18}. However, less focus has been paid to the relational aspects of accessing and using analytics for strategic decision-making and little attention has been paid to senior leaders' readiness to use analytics..."	P5 L25

	We have also made changes to the discussion of this manuscript to position our findings within this wider literature. When discussing relationships between leaders and analysts, we have added the following: “This aligns with previous literature highlighting how NHS leaders with different professional backgrounds can differ in their use of evidence for decision-making and literature emphasising how relationships between evidence producers and users can influence evidence use in UK public health decision-making”. When discussing relationships between leaders of different organisations, we have added the following: “These findings align with previous literature, which reports leader-leader relationships as one of the most important predictors of successful and sustainable partnership working in health and care, as well as a key determinant of evidence-use in NHS and public health decision-making” (see tracked changes).	P14 L20 P15 L5
I also felt it might be worth revisiting in the discussion the theoretical framework about integration mentioned in the introduction, in order to discuss the findings in that context.	We have added the following to the second paragraph of the discussion section of this manuscript to consider our findings in the context of the framework introduced in the introduction: “Our findings support Mowlan and Fulop’s framework by suggesting that greater use of analytics for decision-making may help achieve increased organisational and functional integration. Our findings also suggest that increased integration at organisational and functional levels through joined up data systems could facilitate the use of analytics for informing strategic health and care decision-making⁵.”	P13 L34
There is a minor typo in objectives on p3 - the sentence does not read correctly, is there a word missing?	This has been corrected to “This study investigated the barriers and facilitators that senior leaders’ experience...”	P3 L3

The bullet points under "strengths and limitations" would benefit from being revised slightly. The first bullet simply describes what was done and doesn't clearly articulate the specific strength - is it the timing in relation to ICS/STPs? If so this needs to be more the focus of this sentence. Can this be phrased in terms that make it clearer how this strength makes these findings useful to others? The second point mentions the focus of the study as a digitally engaged site - this is potentially a strength or a limitation; is the strength being flagged here instead the partnership working to inform the study design and conduct?	We have revised the first two bullet points to address these comments and strengthen this section. Bullet point now reads as: "A key strength of this work is that we have illustrated how leaders experience complex and wide-ranging barriers and facilitators of analytics use for strategic decision-making at a time when areas were on the cusp of transitioning from local models of integration in England (Sustainability and Transformation Partnerships) to national statutory organisations (Integrated Care Systems). Our findings are timely as the use of data and analytics are viewed as central to the integration of services and integrated decision-making." Bullet point 2 now reads as: "Another strength is that we worked collaboratively and in partnership with a digitally engaged and innovative site to inform the study design, research questions, study materials and study procedures."	P4 L2-10
In the introduction, the phrase "health and care" is used frequently; I think it would be worthwhile defining what care means in this context, especially for international readers e.g. social care that provides support to people with activities of daily living etc.	We have added the following to the first paragraph to better explain care services: "In England, care services include social care which provides support to those with illness and/or disability with activities of daily life."	P5 L5